# Physical Education and Its Importance to Physical Activity, Vegetable Consumption and Thriving in High School Students in Norway

**DOI:** 10.3390/nu13124432

**Published:** 2021-12-10

**Authors:** Nora Wiium

**Affiliations:** Department of Psychosocial Science, Faculty of Psychology, University of Bergen, 5020 Bergen, Norway; Nora.Wiium@uib.no; Tel.: +47-55-582-849

**Keywords:** physical education, 5Cs of PYD, healthy behaviors, high school students, Norway

## Abstract

Earlier research indicates that physical education (PE) in school is associated with positive outcomes (e.g., healthy lifestyle, psychological well-being, and academic performance). Research assessing associations with resilience and thriving indicators, such as the 5Cs of Positive Youth Development (PYD; *competence*, *confidence*, *character*, *caring*, and *connection*) is limited and more so in the Norwegian context. The aim of the present study was to investigate associations between PE grade (reflecting students’ effort in theoretical and practical aspects of the subject) and the 5Cs as well as healthy behaviors (physical activity (PA), fruit and vegetable consumption), using cross-sectional data collected from 220 high school students in Norway (*M_age_* = 17.30 years old, SD = 1.12; 52% males). Results from structural equation modelling indicated positive associations between PE grade and four of the 5Cs (*competence*, *confidence*, *caring*, and *connection*; standardized coefficient: 0.22–0.60, *p <* 0.05) while in logistic regressions, a unit increase in PE grade was associated with higher likelihood of engaging in PA and vegetable consumption (OR = 1.94; 95% CI = 1.18–3.18 and OR = 1.68; 95% CI = 1.08–2.63, respectively). These significant findings suggest the need for policies and programs that can support effective planning and implementation of PE curriculum. However, further research is needed to probe into the role of PE on youth health and development with representative samples and longitudinal designs.

## 1. Introduction

The positive and protective effects of physical activity (PA), such as enhanced physical health, psychological well-being, increased concentration, academic performance, and reduced feelings of depression and anxiety, have been well documented in earlier studies [1,2,3]. Physical education (PE) is taught as a subject in many countries around the world, but it also incorporates aspects of PA within the school context, because of the different indoor and outdoor activities students engage in during PE sessions. Indeed, Mooses and colleagues [4] found PE to significantly increase daily moderate to vigorous PA alongside reducing sedentary time among schoolchildren. In addition, Tassitano and colleagues [5] observed a positive association between enrollment in PE sessions and several health-related behaviors including physical activity and fruit consumption. 

In many schools, students’ efforts in PE are captured in the grade they receive on the subject. Thus, higher grades in PE would indicate greater efforts and achievement in the physical activities engaged in, which in turn can lead to the promotion of outcomes related to health and development as indicated in earlier studies [1,2,3]. The present study seeks to determine whether this is the case in high school students in Norway.

### 1.1. Physical Education in the Global and Norwegian Contexts

In basic terms, physical education has been described as “education through the physical”. Consistent with United Nations Educational, Scientific and Cultural Organization, PE embraces terms, such as “physical culture”, “movement”, “human motricity”, and “school sport”, and refers to a structured period of directed physical activity in school contexts [6]. A PE curriculum usually features activities such as team and individual games and sports, gymnastics, dance, swimming, outdoor adventure, and track and field athletics [6]. By engaging in a variety of physical activities, students are taught physical, social, mental, and emotional skills to empower them to live an active and healthy lifestyle. PE is also an arena where students can develop and practice skills related to collaboration, communication, creativity, and critical thinking [7]. 

In a world-wide survey of physical education that involved 232 countries (and autonomous regions), 97% of the countries were found to have either legal requirements for PE within their general education systems or PE was a general practice at some ages of the schoolchildren or phases of compulsory schooling [6]. The number of PE lessons that were taught in schools across the countries varied from 0.5 to 6.0 per week and from 16 to 46 weeks per year during compulsory education. Country variation depended greatly on the mindset held about the importance and relevance of the subject in the school curricula.

A European Commission report on physical education and sport at school in Europe indicates that while about 50% of the educational systems have national strategies to support the development of PE and PA, two-thirds have large-scale schemes assigned to these activities [8]. With activities that include athletics, dance, health and fitness, gymnastics, games, outdoor and adventure, swimming, winter sports, and others, the goals of European countries have been to promote the development of pupils and students in the physical, personal, and social domains [8]. 

As in many European countries, PE is one of several subjects taught to pupils and students in compulsory education in Norway (i.e., 6–16-year-olds in primary and lower secondary education). The PE curriculum has both practical and theoretical components. In both components of the curriculum, students are introduced to organized physical activities and spontaneous play in varied environments, in a wide range of sports, dance and other movement activities, and in outdoor life, which allows them to orient and spend time in nature in different seasons as well as being an aspect of exercise and lifestyle that deals with the effect of physical activity on health. In high schools, students receive a total of 168 h of PE lessons during their 3-year education, where in addition to sports activities, outdoor life, and lessons on exercise and lifestyle, they receive education in physical motor activities that go beyond traditional sports activities. Moreover, students at this level of education have the possibility to combine PE with active participation in competitive sports [9]. 

PE lessons in Norway focus on providing students with challenges and courage to enable them to stretch their own boundaries, in both spontaneous and organized activities. In addition, it is anticipated that students will experience joy, mastery and inspiration by participating in a variety of physical activities, which will eventually help them to develop self-esteem, self-understanding, positive perception of the body and positive identity. Furthermore, the social aspects of the physical activities are intended to create an arena where students can exercise fair play and respect for each other [9,10]. All these effects are positive outcomes that tend to signify several components of what has been referred to as the 5Cs of PYD (*competence*, *confidence*, *character*, *caring,* and *connection*) [11] and the ability to develop healthy behaviors, thus supporting health as defined by the World Health Organization. In 1948, the World Health Organization [12] defined health as “a state of complete physical, mental and social well-being and not merely the absence of disease or infirmity” (p. 100). PE’s role in facilitating health and development will thus touch on WHO’s three dimensions of health.

The current focus of the Norwegian PE curriculum is a result of changes made to the curriculum in 2012, due among others to students and teachers’ dissatisfaction with the stress emanating from the expectations attached to sports achievements and physical performance abilities as well as the observation and measurement the teachers had to undertake to grade students’ abilities. With the present curriculum, it is the effort made by students (i.e., the attempts made to use the acquired knowledge and capabilities to reach developmental goals and not necessarily the attained progress) that is considered as relevant [13]. Thus, a high grade in PE subject will not only indicate a form of academic achievement, but it will also signify students’ efforts and experience in a variety of physical activities and their knowledge on how these activities can promote positive developmental outcomes, such as health, self-development, and identity [14].

### 1.2. Positive Youth Development and the 5Cs

Positive Youth Development is a line of research and a developmental framework that focuses on the identification and promotion of youth strengths [15,16], and the equipping of youth toward becoming productive members of their society [16]. PYD suggests that all young people have strengths and as such are potential resources to their own development and that of the society they are a part of. In addition, PYD proposes that all youth contexts, such as home, school and the local community, have human and material resources that youth can have access to in their interactions with significant others in these contexts [17]. In PE sessions, these contextual resources will be the support from peers and teachers, the opportunities created for students to develop resilience, competences and mastery, the boundaries students will have to respect as well as the expectations to be met. Youth strengths will be the personal interest, skills, and abilities that students bring to the PE sessions.

Within PYD, the 5Cs are viewed as a product of the alignment between youth strengths and contextual resources [15]. Accordingly, the dynamic interaction that ensues between an active, engaged, and competent person and their receptive, supportive, and nurturing ecologies in the context of varying degrees of risk and adversity will lead to a process referred to as adaptive developmental regulations [11,15,18], where youth can be resilient, thrive and develop to their full potential. Thriving means youth are scoring high on the 5Cs. The 5Cs include *competence* (which reflects the positive views of an individual’s action in domains, such as academic, social, cognitive and vocational); *confidence* (which relates to the individual’s sense of mastery and purpose for the future, a positive identity and self-efficacy); *character* (which denotes one’s integrity, moral commitment, and respect for societal and cultural rules); *caring* (which indicates one’s sense of empathy and sympathy for others); and *connection* (which reflects the bidirectional exchanges and healthy relations between the individual and friends, family, school, and community). Within the PYD framework, the 5Cs typically reflect thriving and positive development, but also resilience (in contexts where there are high levels of risk and adversity) among young people [15,18]. PYD proposes that youth who are thriving are put on a life trajectory towards an “idealized adulthood” [19]. In addition, youth who are resilient and thriving are more likely to contribute to their own development as well as to the development of their society [15].

### 1.3. Earlier Research on Positive Youth Development, Healthy Behaviors and Physical Education

Research on the relationship between grade in PE (which reflects students’ participation in PE sessions) and the 5Cs of PYD is limited, although earlier studies have recounted several positive outcomes of PE in schools. In one study that investigated PYD-related outcomes in the contexts of PE, Holt et al. [20] found in a qualitative study of 8 teachers and 59 children at an inner-city school in Canada that PE activities engaged in tended to promote developmental outcomes, such as empathy and healthy relationships between students. In addition, PE activities became an arena where teachers considered students’ input to the PE activities and created boundaries and procedures for expected behaviors.

Furthermore, Bailey [1], in a review article, summarized several positive and profound benefits of PE that included physical health, healthy lifestyle, psychological well-being, social skills and improved academic performance. These benefits were more probable in contexts where there were positive experiences of the PE activities, enjoyment, efforts made to engage all students as well as when teachers and coaches were committed and were equipped with the necessary skills. In another literature review on the impact of PE and sport on educational outcomes, Stead and Nevill [21] found that increased physical education, physical activity or sport tended to maintain or enhance academic achievement. The authors also found a positive association between physical activity and aspects of mental health, such as self-esteem, emotive well-being, spirituality, and future expectations. Moreover, Stead and Nevill [21] observed that the implementation of extra organized physical activity, as little as 10 min into the school day, tended to improve classroom behavior. These earlier studies support the important role of PE on health (including the physical, mental, and social dimensions) and positive development in youth.

As for healthy behaviors and their associations with PE, Mayorga-Vega and colleagues [22] conducted a study among 158 students in a Spanish high school and found that students had greater physical activity levels and lower levels of sedentary behaviors during PE days compared to non-PE days and weekends. In a much larger sample of 4210 high school students in Brazil, Tassitano et al.’s [5] assessment of the role of PE enrollment on several health behaviors revealed, among others, positive associations of enrollment in PE classes with physical activity and fruit consumption, as well as a negative association with drinking of sugar-sweetened beverages. In a longitudinal study of Canadian adolescents, Wiseman and Weir [23] investigated PE rating among other subjects alongside the importance of PE for PA levels and several health variables over a two-year period. Their results indicated that most of the participants (78%) preferred PE over other subjects, and that preferring PE was associated with higher PA levels, lower BMI, and higher self-esteem. Thus, while earlier research supports the predictive role of PE on youth development and healthy behaviors, the evidence regarding the importance of PE to the 5Cs of PYD is unclear because of limited research.

### 1.4. Aims of the Present Study

Research on the 5Cs of PYD has usually involved American youth [11,24] although research featuring non-American samples is growing [25,26]. Moreover, while the effects of PE on youth health and development have been widely studied, a literature search did not return any study that had assessed the relation between PE and the 5Cs in the Norwegian context. Several studies have hinted how activities engaged in during PE can be used to foster positive development. For example, Mandigo et al. [27] described how quality PE activities can be used to promote positive development and peace education among schoolchildren in a developing country. More specifically, the authors outlined various behaviors in the physical, intellectual, psychological, and social domains that physical educators can instill in schoolchildren to foster the 5Cs of PYD and peaceful interactions. Holt and colleagues [20] also described how strategies, such as setting of clear boundaries and allowing inputs from schoolchildren, and the teacher being a PE specialist, could facilitate positive youth development. Thus, in line with these earlier PYD studies, PE can be an arena where youth development as well as health (as proposed by WHO) are promoted. 

In the present study, the aim is to examine the link between grade in PE and positive outcomes reflected in the 5Cs of PYD. A second aim is to study the association between PE grade and healthy behaviors, such as PA during leisure time and the consumption of fruit and vegetables. With the goal of the Norwegian PE curriculum to promote health, self-development and identity among others, grade in PE reflecting attained knowledge, participation and efforts invested in various physical activities should be associated with the 5Cs. Thus, as a hypothesis, students with higher PE grades are also expected to report higher scores on the 5Cs. Like the 5Cs, positive associations are hypothesized between PE grade and healthy behaviors. If positive associations are found between PE, the 5Cs and healthy behaviors, PE can be considered as an avenue to instill competencies that can have implications for students’ health, thriving, and resilience. Earlier studies suggest that boys engage in PA more often than girls, and PA tends to decrease with age [28]. Parents’ educational level has also been found to be positively related to the 5Cs [26]. Hence, gender, age, and parents’ education were accounted for in the assessment of the influence of PE grade on the 5Cs and healthy behaviors.

## 2. Materials and Methods

### 2.1. Sample

The current study forms part of a larger international project on positive development among youth and emerging adults, where the general goal is to assess how youth strengths and contextual resources align to foster thriving and youth contribution to societal development [29]. For the present study, cross-sectional data were collected from 220 students in four high schools located in Eastern and Western Norway. About 52% of the participants were boys and the age range was between 16 and 20 years (*M* = 17.30, SD = 1.12). Almost 83% reported that the highest level of education of their father was postsecondary, while 87% did the same for their mother’s education.

### 2.2. Measures

#### 2.2.1. Physical Education Grade

Participants self-reported their current academic grade (1 to 6) on physical education. A grade of 1 represents minimum knowledge and effort invested during PE sessions while a grade of 6 represents great knowledge and maximum invested effort in PE sessions.

#### 2.2.2. The 5Cs of PYD

To assess the 5Cs, Geldhof and colleagues’ [11] short version of the PYD questionnaire, consisting of 34 items, was used. Samples of the items used in measuring the 5Cs include: “I am just as smart as others my age” (*competence*, 6 items); “I really like the way I look” (*confidence*, 6 items); “I usually act the way I am supposed to” (*character*, 8 items); “When I see someone being exploited I want to help them” (*caring*, 6 items); and “I am a helpful and important family member” (*connection*, 8 items). Responses were measured on a 5-point Likert scale, ranging from 1 (Strongly Disagree) to 5 (Strongly Agree), for example, where a higher score indicated a higher experience of the C-item in question. The psychometric properties of the 5Cs scale have been mostly assessed in U.S. samples [11,24] but also in some non-U.S. samples [25,26].

#### 2.2.3. Healthy Behaviors

Items measuring healthy behaviors (physical activity, fruit and vegetable consumption) were adopted from the Search Institute’s [30] survey on attitudes and behaviors. Participants indicated 0 (No) or 1 (Yes) to the following items: “I engage in physical activity (for at least 30 min) twice or more per week”, “I eat at least one serving of fruit every day” and “I eat at least one serving of vegetables every day”. Spearman correlation among the three healthy behaviors ranged from 0.25 to 0.37.

#### 2.2.4. Demographic Variables

Data were also collected on gender (boy or girl), age and mother and father’s educational level (five levels of education: 1 (no education), 2 (primary school), 3 (high school), 4 (technical or vocational school), and 5 (university)). The demographics were treated as control variables in the data analysis. 

### 2.3. Procedure

Data collection took place in May–August 2019. Convenience sampling was used to select four schools located in the Eastern and Western parts of Norway. The heads of the conveniently selected schools were contacted via e-mail, with a request to participate in the study and an information letter about the purpose of the study. After agreeing to participate, the heads of schools were sent informed consent forms, developed in accordance with the NSD (Norwegian Centre for Research Data) guidelines, which they were asked to sign and send back. Once that was done, teachers from the four schools who agreed to conduct the survey with their students were sent the questionnaire via email. Informed consent was sought from students prior to the data collection, which took place during school hours over the schools’ internal web system. NSD (Norwegian Centre for Research Data) approved the study (51708/3/IJJ), while Semantix Translations Norway AS, Oslo, Norway, a company that specializes in interpretation services, translated the questionnaire from English to Norwegian using double-checking methods and translation experts in the relevant field of research to ensure preservation of meaning.

### 2.4. Data Analysis

G*Power 3 [31] was used to conduct a power analysis to determine the sample size that will allow for the assessment of meaningful associations and the detection of effect sizes (small, medium, or large). Using a two-tailed test with the 5 independent variables (PE grade and the four demographic variables (gender, age, father’s education and mother’s education)), and an alpha value of 0.05, the results indicated that with a power of 0.80, sample sizes of 395, 55, and 25 were needed to detect effect sizes of 0.02 (small), 0.15 (medium), and 0.35 (large), respectively. Reaching the study’s sample size of 220 meant that medium to large effect sizes can be detected in the statistical analyses. 

Descriptive and correlation analyses were performed using IBM SPSS Statistics for Windows, version 25, while all other analyses were carried out using Mplus version 8 [32]. Most participants (80%) were missing only 3 cases or less, while 59% had full data. The analyses in Mplus were conducted with the Maximum likelihood estimation, an estimation method used to handle missing cases. The method works by estimating a likelihood function for each case based on the variables present in the dataset such that all the available data are used.

Descriptive analyses were conducted to assess the pattern of study variables: the demographics, PE grade, the 5Cs of PYD and the three healthy behaviors. Confirmatory factor analysis (CFA) was performed on the items measuring the 5Cs to verify the factorial structure of the scale. Chi-square tests and indices, such as the Tucker Lewis Index (TLI; acceptable above 0.90), the Root Mean Square Error of Approximation (RMSEA; acceptable below 0.08), and Comparative Fit Index (CFI; acceptable above 0.90) [33,34]) were used to evaluate model fit. To test the hypothesis that higher scores in PE will be associated with higher scores in the 5Cs, structural equation modelling (SEM) analysis was carried out. In preliminary analyses, the linearity and normal distribution of the 5Cs as dependent variables were determined, with skewness and kurtosis falling within the acceptable range of −2 to +2 and −7 to +7, respectively for SEM analysis [35]. Finally, the hypothesis that higher scores in PE will be associated with higher odds of the healthy behaviors was tested using logistic regressions due to the binary response categories of the healthy behavior variables. In both SEM and logistic regression, the demographic variables: gender, age, and father’s and mother’s educational background were controlled for.

## 3. Results

### 3.1. Descriptive Analysis

In Table 1, a frequency analysis of PE grade showed that about 96% of the participants reported grades between 4 and 6. In the Norwegian high school system, a grade of 1 is the lowest, while 6 is the highest a student can earn in a subject. For the 5Cs of PYD, high Cronbach’s alphas, indicating high internal consistencies (ranging from 0.85–0.93) were estimated for all the Cs. The frequency distribution of the three healthy behaviors revealed that most of the participants (about 82%) engaged in PA for at least 30 min twice or more per week, while 57% and 70% consumed at least one serving of fruit and vegetable per day, respectively (Table 1).

Furthermore, descriptive analysis of the 5Cs showed that the highest mean score was registered for *caring* (*M* = 4.29, SD = 0.78), followed by *character* and then *connection*. *Competence* had the lowest mean score (*M* = 3.65, SD = 0.86). Thus, on average, participants’ responses on the 5Cs suggested moderate to relatively high levels of the PYD outcomes. The statistically significant correlations between PE grade and the 5Cs (mean scores) were weak to moderate, ranging from 0.17 to 0.55. In addition, the correlation between PE grade and the healthy behaviors were weak but statistically significant (0.19–0.25). Finally, several significant but weak correlations were observed between the 5Cs and the healthy behavior variables as well as between the demographic variables, the 5Cs and the healthy behavior variables (Table 2).

### 3.2. CFA of the 5Cs of PYD and Structural Equation Modelling of PE Grade and the 5Cs

Prior to the assessment of the associations between PE grade and the 5Cs, confirmatory factor analysis (CFA) was conducted on the 34 items of the 5Cs to determine the factorial structure of the scale. An initial CFA of the items, where 14 pairs of same-facet items (in *competence*, *confidence*, *character* and *connection*) were allowed to correlate, yielded a poor model fit: *χ^2^* (500, *N* = 194) = 998.075, *p* < 0.001, *RMSEA* = 0.072, *CFI* = 0.872, *TFI* = 0.857. An examination of the modification indices revealed cross-loadings of four items, two items regarding social competence for *competence*, one item on social conscience for *character* and another on *caring*. In addition, the modification indices indicated correlations among one pair of same-construct items (i.e., *confidence*) and two pairs of different-construct items, one between *competence* and *connection*, and the other between *confidence* and *character*. After eliminating cross-loading items and including the correlations, an adequate model fit was attained in a second CFA: *χ^2^* (378, *N* = 194) = 646.879, *p* < 0.001, *RMSEA* = 0.061, *CFI* = 0.917, *TFI* = 0.905. The factor loadings for all 5Cs in this new CFA were adequate, ranging from 0.54 to 0.91. Correlations among the latent factors of the 5Cs were between 0.32 and 0.88.

In Table 3, having controlled for demographic factors (i.e., gender, age, and parents’ educational background), findings from the structural equation modelling revealed significant associations between PE grade and all the 5Cs of PYD except for *character*. Not surprisingly, the strongest association was between PE grade and *competence* (standardized coefficient of 0.60), both largely reflecting students’ competence. The standardized coefficients for *confidence* and *connection* were 0.36, and 0.37, respectively, while for *caring* the coefficient was 0.22. Thus, higher scores in PE were significantly associated with higher scores in the 5Cs besides *character*. As for the demographic variables, only gender was significantly related to *caring* in the SEM analysis (standardized coefficient of 0.36), where girls scored higher than boys.

### 3.3. Logistic Regression Analyses of Physical Education and Healthy Behaviors

For the associations between PE grade and healthy behaviors, logistic regression models were analyzed because of the binary response categories of the behaviors (Table 4). After controlling for the demographic variables, a unit increase in PE grade was associated with a 94% higher likelihood of engaging in PA (OR = 1.94; 95% CI = 1.18–3.18), and a 68% higher likelihood of vegetable consumption (OR = 1.68; 95% CI = 1.08–2.63), that is, when all other variables in the models were held at a constant. Thus, PE grade was significantly related to higher odds of PA and vegetable consumption, while the association with fruit consumption was not significant. None of the demographic variables were significantly related to the healthy behavior variables in the logistic regression analyses.

## 4. Discussion

The aim of the present study was to investigate the associations of PE grade with the 5Cs of PYD and healthy behaviors. As hypothesized, positive associations were observed between PE grade and four of the 5Cs (*competence*, *confidence*, *caring*, and *connection*) after adjusting for gender, age, and father’s and mother’s educational background. In contrast, although there was an indication that *character* was associated with PE grade, this association was not statistically significant in the SEM analysis. For the associations between PE grade and healthy behaviors, while logistic regression analyses showed higher odds of engagement in PA and vegetable consumption with every unit increase in PE grade, no such association was found for fruit consumption. Thus, the hypotheses were confirmed, although not for the association of PE grade with *character* and fruit consumption. That PE was found to be largely associated with the 5Cs and healthy behaviors is consistent with earlier findings that have supported the significant role of PE sessions on positive outcomes reflecting WHO’s different dimensions of health (physical, mental, and social) [1,21].

The current finding that PE grade was strongly related to *competence* was no surprise, as both connote a form of academic competence. In the present study, *competence* as one of the 5Cs was measured as competence in the academic and physical domains. Thus, PE grade was not only related to academic competence or cognitive abilities, but also to physical competence in sports and athletic activities. Earlier research among German students that supports the current findings reported a positive association between PE and cognitive skills measured by grades in German and mathematics [36], while findings of a review article also indicated that increasing the amount of time dedicated to PE and sports was in many instances associated with academic performance [1]. The goal of the Norwegian PE curriculum to enable students to develop mastery in the skills needed to undertake a variety of physical activities [9] can therefore be important not just for the grade in PE but for the general academic competence of students as well.

In addition to being associated with *competence*, PE grade was associated with *confidence*, *caring* and *connection.* Accordingly, students who scored high in PE were also more likely to report indicators of thriving and positive development, associations that have been confirmed in a related study on the link between participation in sport camps and the 5Cs of PYD that were captured as two factors (pro-social values and confidence/competence) [37]. Moreover, Bailey [1] in a review, reported on how PE and sports in schools can provide a favorable environment for social development, a finding that largely corroborates the current results on the significant link between PE and *connection* (signifying healthy social relations at home, school, and local community). Indeed, an important aim of the Norwegian PE curriculum among others is to create a social arena for fair play and respect between students [9,10]. However, *character* (reflecting the integrity and moral compass of youth) was the only thriving indicator that was not associated with PE grade, neither in zero-order correlation nor in multivariate analysis. It is possible that the alignment between youth strengths and contextual resources that facilitate the 5Cs of PYD in PE sessions predicts some of the Cs better than others. This assertion will need to be probed into in future research.

Furthermore, PE grade was related to healthy behaviors, such as PA and vegetable consumption, but not fruit consumption. Earlier research among students attending a Spanish high school associated participation in PE with greater PA levels and lower levels of sedentary behaviors during PE days compared to non-PE days and weekends [22]. Enrollment in PE activities among high school students in Brazil has been found to be positively related to healthy behaviors, such as PA and fruit consumption, as well as negatively related to drinking of sugar-sweetened beverages [5]. Wiseman and Weir [23] also found among Canadian high school students that preferring PE over other school subjects was associated with higher PA levels, lower BMI, and higher self-esteem. Although it was PE grade that was assessed in the current study, the grade reflects students’ participation in both theoretical and practical components of the Norwegian PE sessions. Thus, the current finding on the positive association between PE grade and healthy behaviors is largely in line with earlier findings. In summary, PE sessions reflected in the grade of students were associated with positive youth developmental outcomes, such as thriving (the 5Cs) and healthy behaviors, outcomes that tend to reflect all three dimensions of health (physical, mental, and social) as defined by the World Health Organization.

In SEM and logistic regression, the demographics did not appear to play an important role on the 5Cs and healthy behaviors, as a significant association was only observed between gender and *caring*, with girls reporting higher scores than boys. This finding is in line with earlier research that found similar associations in upper secondary and university students in Spain [38] and is often attributed to gender socialization, where boys are taught to be tough and girls caring. In future studies, the role of gender and other demographics are worth investigating to ascertain their effects and place in intervention programs.

### 4.1. Limitations

The present study has some limitations that need to be considered in the interpretation of the findings. First, the relationships between PE grade and the positive youth developmental outcomes may not indicate causation due to the cross-sectional design of the current study. While the present and earlier findings suggest a positive influence of PE on youth development and healthy behaviors, it is also possible that high levels of the thriving indicators (*competence*, *confidence*, *caring* and *connection*) led to more effort in PE sessions, and consequently, high grade in the subject. In addition, it is likely that students who participate in healthy behaviors such as PA and vegetable consumption will also perform better in PE sessions. Looking at these relationships within a longitudinal design will shed more light on both the developmental trajectories and relations between PE participation and positive youth outcomes.

Second, while there is no reason to believe that youth will be deceptive in the report of their grade and competencies, it is still likely that their self-report responses were affected by social desirability bias, where they tended to over-report their PE grades, for example. In future studies, students’ actual grades provided by teachers can be one method to address the limitation associated with self-report responses and the associated social desirability bias. Third, the binary response categories (Yes/No) of the healthy behaviors did not allow much variation among the behaviors to be assessed. Moreover, although the measures represented general assessment of PA and fruit and vegetable consumption, they did not adequately reflect the global recommendations of the healthy behaviors. This is a limitation that can be addressed in future studies with better instruments that allow for more variations as well as assessment of the recommended amounts and levels of the healthy behaviors. Fourth, the items measuring the 5Cs of PYD were created with US samples, and although the scale was largely validated with the Norwegian sample, there were some items that cross-loaded onto different factors. In addition, relatively high correlations were found among some of the measures, for example between *competence* and *confidence*. Thus, it is possible that some items of the 5Cs did not adequately capture or make a distinction between the thriving indicators in Norwegian students. These shortcomings can be a topic of investigation in future studies using qualitative methods.

Finally, although the power analysis indicated that the sample size of 220 was enough to detect medium to large effect sizes in the relationships being studied, a larger sample could provide more robust findings. Besides, the participating schools and thus the students involved in the current study were selected through convenience sampling, thus limiting the extent to which the present findings can be generalized to the whole youth population in Norway. Future studies that use a more representative and inclusive sample reflecting youth from different geographic locations, diverse ethnicities and other backgrounds will be more effective in generating findings that are representative of the Norwegian youth population.

### 4.2. Implications for Research, Policy, and Practice

Despite the limitations, the current study has implications for research, policy, and practice. In terms of research, the validation of the 5Cs of PYD scale among high school students in Norway adds to the limited research of the 5Cs in Norway and paves the way for further research of the thriving indicators among youth in the Norwegian and other similar Scandinavian and European contexts. Additional research on the 5Cs can also eventually lead to a more refined scale that includes items unique to the Norwegian, Scandinavian or European context. In addition, future studies on PE and the 5Cs can assess the level of risk and adversity in the contexts in which youth are interacting. This will enable the assessment of not only thriving, but resilience as well.

As for policy, the fact that PE grade is related to thriving and healthy behaviors suggests that the Norwegian PE curriculum is important to the promotion of the positive development of the youth, and, possibly, resilience. These results should make the effective implementation of PE curriculum in all schools a priority on the Norwegian political agenda at both the national and community or school level. This way, young people across gender, socio-economic statuses, ethnicities, and other backgrounds can be reached and empowered with the necessary physical, cognitive, and psychosocial skills and competences that are associated with the array of activities taught in PE sessions. Moreover, the current findings of the significant role of PE can inform strategies used in PE curricula in other Scandinavian and European countries. In line with a European Commission report [8], although all European countries acknowledge the importance of PE at school, only two-thirds of the educational systems had large-scale national initiatives to support the promotion of PE and PA. Indeed, as implied in the current findings, the goal of European countries to facilitate the physical, personal and social development of pupils and students can only be realized when PE curricula are planned and implemented effectively. 

There are some practical implications of the current findings as well. With the significant associations between PE grade, the 5Cs of PYD (indicating thriving indicators), and healthy behaviors, it is important that during PE sessions, efforts are made to engage all students in activities that can create positive experiences, enjoyment and mastery as outlined in the PE curriculum. In the curriculum, there is also a focus to provide students with challenges that can enable them to participate actively in both spontaneous and organized activities as well as arenas where students can exercise fair play and respect for each other. Efforts made to implement all these aims in the PE sessions will not only produce healthy, thriving, and resilient youth but, as proposed by PYD, the efforts would also mean a healthy transition into adulthood for the youth. 

## 5. Conclusions

Positive effects of PE participation have been well documented in earlier studies. The current study adds to these benefits with findings that suggest that PE grade reflecting participation in PE is significantly related to thriving indicators, such as *competence*, *confidence*, *caring* and *connection* (4 out of the 5Cs of PYD), as well as healthy behaviors such as PA and vegetable consumption. These findings support the importance of PE sessions to the healthy development of youth and suggest that policies and programs at the national and local levels that ensure the effective implementation of a PE curriculum in school would be promoting developmental outcomes that align with the dimensions of health outlined by the World Health Organization. However, more research needs to be carried out with adequate measurement of healthy behaviors and representative samples to ascertain the facilitating role of PE sessions on youth health, thriving, and positive development, but also resilience in risk and adverse contexts of youth, as this can secure a life trajectory towards an idealized adulthood for all youth.

## Figures and Tables

**Table 1 nutrients-13-04432-t001:** Descriptive statistics and reliability coefficients for study variables among Norwegian youth.

Study Variables		*N* = 220
Gender (%)		
	Male	52.3
	Female	47.7
Age (%)		
	16	31.0
	17	25.9
	18	28.2
	19	11.6
	20	3.2
Father’s education (%)		
	High school or lower	16.7
	More than high school	83.3
Mother’s education (%)		
	High school or lower	12.6
	More than high school	87.4
Grade in Physical Education (%)		
	1—Lowest grade	1.1
	2	0.6
	3	2.8
	4	26.5
	5	52.5
	6—Highest grade	16.6
PYD Measures (Cronbach’s alpha—α)		
	Competence	0.88
	Confidence	0.86
	Character	0.85
	Caring	0.89
	Connection	0.93
Healthy behaviours (%)		
	Physical activity—(for at least 30 min) twice or more per week	81.9
	Fruit consumption—at least one serving a day	57.4
	Vegetable consumption—at least one serving a day	69.9

**Table 2 nutrients-13-04432-t002:** Correlation analyses of demographic variables, physical education grade, the 5Cs of PYD, and healthy behaviors.

Study Variables	2	3	4	5	6	7	8	9	10	11	12	13
1. Gender	−0.02	−0.12	−0.10	−0.13	−0.21 **	−0.18 *	0.07	0.28 **	−0.01	0.06	−0.04	0.08
2. Age	-	−0.20 **	−0.17*	−0.08	−0.14	−0.12	−0.06	−0.05	−0.15 *	−0.13	−0.09	−0.08
3. Father’s education		-	0.38 **	0.06	0.20 *	0.15	0.08	−0.03	0.15 *	0.03	0.09	0.12
4. Mother’s education			-	0.16 *	0.27 **	0.19 *	0.14	0.11	0.16 *	0.05	−0.01	0.05
5. Physical education				-	0.55 **	0.38 **	0.12	0.17 *	0.29 **	0.25 **	0.20 **	0.19 *
6. Competence					-	0.78 **	0.53 **	0.35 **	0.72 **	0.25 **	0.28 **	0.25 **
7. Confidence						-	0.65 **	0.33 **	0.68 **	0.22 **	0.15 *	0.15 *
8. Character							-	0.66 **	0.65 **	0.10	0.13	0.14
9. Caring								-	0.48 **	0.16 *	0.06	0.11
10. Connection									-	0.16 *	0.21 **	0.18 *
11. Physical activity										-	0.25 **	0.33 **
12. Fruit consumption											-	0.37 **
13. Vegetable consumption												-
Descriptive analysis												
Range	16–20	1–5	1–5	1–6	1–5	1–5	1–5	1–5	1–5	0–1	0–1	0–1
Mean (SD)	17.30 (1.12)	4.40 (0.88)	4.58 (0.88)	4.78 (0.85)	3.65 (0.86)	3.73 (0.97)	3.94 (0.96)	4.29 (0.78)	3.82 (0.77)	0.82 (0.39)	0.57 (0.50)	0.70 (0.46)

Note. * *p* < 0.05; ** *p* < 0.01.

**Table 3 nutrients-13-04432-t003:** Structural equation model of physical education grade and the 5Cs of PYD.

PE Grade ^a^	5Cs of PYD				
	Competence	Confidence	Character	Caring	Connection
Estimate *	0.60	0.36	0.19	0.22	0.37
S.E.	0.13	0.10	0.12	0.10	0.12
Est./S.E.	4.58	3.55	1.53	2.14	3.11
*p* value	0.000	0.000	0.126	0.032	0.002

Note. PE—Physical education; ^a^ Controlled for gender, age, father’s education and mother’s education; * Standardized coefficient. Italics and bold show significant levels less than 0.05.

**Table 4 nutrients-13-04432-t004:** Associations between physical education (PE) and healthy behaviours: logistic regression analysis.

	Physical Activity					Fruit Consumption					Vegetable Consumption				
	B	S.E.	Sig	OR	95% CI	B	S.E.	Sig	OR	95% CI	B	S.E.	Sig	OR	95% CI
Demographic variables Gender	0.64	0.47	0.180	1.89	0.75–4.78	−0.11	0.35	0.748	0.89	0.45–1.78	0.72	0.40	0.067	2.06	0.95–4.48
Age	−0.13	0.21	0.532	0.88	0.59–1.31	−0.23	0.16	0.149	0.80	0.58–1.09	−0.16	0.17	0.342	0.85	0.61–1.19
Father’s education	0.14	0.29	0.623	1.15	0.66–2.02	0.26	0.22	0.224	1.31	0.85–2.01	0.28	0.23	0.231	1.32	0.84–2.09
Mother’s education	−0.07	0.30	0.815	0.93	0.52–1.68	−0.16	0.23	0.485	0.85	0.55–1.33	0.10	0.24	0.672	1.11	0.70–1.76
Predictor PE grade	0.66	0.25	0.009	1.94	1.18–3.18	0.38	0.21	0.070	1.46	0.97–2.20	0.52	0.23	0.022	1.68	1.08–2.63

Note. PE—Physical education; B—Unstandardized coefficient; S.E.—Standard Error; Sig—Significance level; OR—Odds Ratio; CI—Confidence Interval.

## Data Availability

Data supporting reported results can be found on the following link: https://teams.microsoft.com/_#/school/files/General?threadId=19%3A00faa60f3ab64020b836a1c964c56962%40thread.skype&ctx=channel&context=PYD%2520Database&rootfolder=%252Fsites%252FTEAM_PYDCrossNational_Project%252FShared%2520Documents%252FGeneral%252FPYD%2520Database (accessed on 18 August 2021).

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
