# Peer review of "Physical Education and Its Importance to Physical Activity, Vegetable Consumption and Thriving in High School Students in Norway"

_nutrients, 2021, doi:10.3390/nu13124432_

Round 1

Reviewer 1 Report

The manuscript entitled ‘Physical Education and its Importance to Physical Activity, Vegetable Consumption and Thriving in High School Students in Norway’ presents interesting issue, however some corrections are needed

  • Line 15 – ‘M =17.30, SD = 1.12; 52% males).’ – please add ‘year old’
  • Please add specific numeric data accompanied by p-Values in the abstract.
  • Please add some conclusions (not only statements reproducing results) at the end of the abstract.
  • In this section (1.1. Positive Youth Development, Healthy behaviors and physical education) Author presented the information associated with the Positive Youth Development. This section should be presented – what do we know and what is the background for this study. Some detailed information about other studies are necessary (international context – the situation in other countries should be presented as in lines 155-167 – this section should be widened). The good background should present the history of problem, the current knowledge and scientific "gap", and then authors should present how their study could fill this gap to justify the study.
  • The title ‘1.3. The Present Study’ – is not very communicative
  • Methodology: ‘’ Items measuring healthy behaviors (physical activity, fruit intake, and vegetable consumption)” – please add more detailed information about fruit intake, and vegetable intake (the serving size was characterized?). (It is consumption or intake – please be consisted)
  • the major issues is associated with Fruit and Vegetable consumption – simple question is not a valid tool. Moreover, the information that it is at least one serving a day give almost no information (if recommendation of F&V intake are meet? )
  • For the research that involves human subjects the rules of the  Declaration of Helsinki of 1975 must be applied, including ethics commission approval and especially informed consent. Please add the information about number of ethics commission approval (specific reference)
  • Was the normality of distribution tested? The information about it should be added and authors should be consequent. If data have normal distribution, they should be treated as such, if not, nonparametric tests should be applied. Please specify it.
  • In discussion section – please do not presented the data (do not presented in table or figure)
  • Conclusion ‘Positive effects of PE participation, such as higher levels of PA, fruit consumption, lower BMI, and higher self-esteem have been reported in earlier studies. However, in the Norwegian context, limited research has been done on positive outcomes or thriving indicators, such as the 5Cs of PYD and their associations with PE in schools. The current study investigated these associations as well as the influence on healthy behaviors, such as PA, fruit and vegetable consumption and found that PE grade, which captures students’ achievement in both theoretical and practical aspects of the subject, was related to thriving and engagement in healthy behaviors.’ – this paragraph is irrelevant. Please present only the most important (prominent) concussion
  •  

Author Response

Comments and Suggestions for Authors

The manuscript entitled ‘Physical Education and its Importance to Physical Activity, Vegetable Consumption and Thriving in High School Students in Norway’ presents interesting issue, however some corrections are needed

Line 15 – ‘M =17.30, SD = 1.12; 52% males).’ – please add ‘year old’

RESPONSE: Thank you for the comment, ‘year old’ is added now.

Please add specific numeric data accompanied by p-Values in the abstract.

RESPONSE: This comment has been addressed. Please see abstract.

Please add some conclusions (not only statements reproducing results) at the end of the abstract.

RESPONSE: The conclusions have been made clearer in the revision.

In this section (1.1. Positive Youth Development, Healthy behaviors and physical education) Author presented the information associated with the Positive Youth Development. This section should be presented – what do we know and what is the background for this study. Some detailed information about other studies are necessary (international context – the situation in other countries should be presented as in lines 155-167 – this section should be widened). The good background should present the history of problem, the current knowledge and scientific "gap", and then authors should present how their study could fill this gap to justify the study.

RESPONSE: Thank you for the comments on the introduction. In the revision the issues raised have been addressed. Please see the introduction section.

The title ‘1.3. The Present Study’ – is not very communicative

RESPONSE: The title has been revised to make it more communicative. See page 4

Methodology: ‘’ Items measuring healthy behaviors (physical activity, fruit intake, and vegetable consumption)” – please add more detailed information about fruit intake, and vegetable intake (the serving size was characterized?). (It is consumption or intake – please be consisted)

the major issues is associated with Fruit and Vegetable consumption – simple question is not a valid tool. Moreover, the information that it is at least one serving a day give almost no information (if recommendation of F&V intake are meet? )

RESPONSE: Available information on the healthy behaviors was presented. I have acknowledged these measurements as not optimal and not reflecting the recommendations. I have accordingly discussed the related limitations. Please see page 13 for the limitation section.  Also “consumption” has been used more consistently throughout the manuscript.

For the research that involves human subjects the rules of the  Declaration of Helsinki of 1975 must be applied, including ethics commission approval and especially informed consent. Please add the information about number of ethics commission approval (specific reference)

RESPONSE: Thank you for the comment. All the ethical issues raised can be found in the “Procedure” section on page 5.

Was the normality of distribution tested? The information about it should be added and authors should be consequent. If data have normal distribution, they should be treated as such, if not, nonparametric tests should be applied. Please specify it.

RESPONSE: In the SEM analysis, the 5Cs were examined as the dependent variables. Information that a test of normality of distribution was performed has now been included. See page 6.

In discussion section – please do not presented the data (do not presented in table or figure)

RESPONSE: All the tables are now in the results section.

Conclusion ‘Positive effects of PE participation, such as higher levels of PA, fruit consumption, lower BMI, and higher self-esteem have been reported in earlier studies. However, in the Norwegian context, limited research has been done on positive outcomes or thriving indicators, such as the 5Cs of PYD and their associations with PE in schools. The current study investigated these associations as well as the influence on healthy behaviors, such as PA, fruit and vegetable consumption and found that PE grade, which captures students’ achievement in both theoretical and practical aspects of the subject, was related to thriving and engagement in healthy behaviors.’ – this paragraph is irrelevant. Please present only the most important (prominent) concussion

RESPONSE: The conclusions have now been revised to make it more focused. Please see page 13.

Reviewer 2 Report

Physical Education and its Importance to Physical Activity, Vegetable Consumption and Thriving in High School Students in Norway

            This study contributes to the body of knowledge about physical education and its benefits for the youngsters. This study findings are parallel to many other studies from all around the world, with standardised procedures and statistics. The main advantage of this study is well performed statistical analysis with up-to-date tools and computation. Authors have enough courage to properly state its limitations, but this does not justify lack of insight into the context of Norwegian curriculum on more specific level or the demographic context of the participants and their families. Methodologically, introduction needs to be rearranged to make the reasoning more straightforward and in order. Specific concerns are addressed below.

Comments

Abstract

Line 15 – Is M a mean age of participants?

Key values of statistical results could be put in numbers to highlight strength of evidence, especially for main topic, which is association between PE and Vegetable Consumption or Thirving.

Introduction

The quality of content and most of explanation are sufficient. What needs to be changed is a structure. The order of the introduction/background section is to introduce the reader to the topic, present premises which leads to certain assumptions. Then hypothesis is formed and aim of the research should be stated at the end of this section.

Present structure of:

  1. Introduction

1.1. Physical education in the global and Norwegian contexts

1.2. Positive Youth Development, Healthy behaviors, and physical education

1.3. The Present Study

Should be rearranged to:

  1. Introduction without stating the aim – just general presentation of topic with basic definitions. Context of PE in the world and Norway could be state here (from 1.1)
  2. Clear description of PYD with exact explanation of each C. There is a place to do so, and reader will skip it than go to references with casual reading attitude.
  3. Presentation of the current body of knowledge and its limitations, forming hypothesis and a reason to do this research, finally stating aim here.

Methods

Line 205 – did the scores were self-reported or it was backed by their school grades and PE teacher evaluation?

Line 217, Line 225 – if this is a result of this study, the information should be put into results section

Line 227 – Please add information that parent’s education level was put in the number scale accordingly from 1 to 5. There is also flaw here. There are 220 students. All students have two parents or caretakers in traditional heterosexual family? There is a chance that someone live only with one parent, or someone was adopted. Without clarification, this is a bias of this study and for group that large and randomly chosen students there is unlikely that all families present one model – maybe the lack of data in some places is due to this issue here.

Line 239 – some of students were 16 years old. Is Norwegian law treating them as adults or consent ought to be obtained from their care takers/parents?

Line 241 – I understand that this external service performed language validation procedure. Do you have α-Cronbach values for respective items or other statistics that shows consistency with original version?

Line 249 – Please state chosen demographic variables.

Line 255 – How many students fulfil the data in 100% without necessity to fill in gaps? This will show exact extent of estimation that was performed.

Line 265-270 – Those hypotheses should be also placed in the introduction.

Results

Please work on the aesthetics in terms of symmetry in all tables. For example, there is no need to split names in table 2 and column 1 in table 1 can be adjusted to one side or central.

Table 3 and 4 should be moved from discussion to result section.

Discussion

Line 330-340 – This is repetition of results. In discussion you should try to compare results to a previous study that you mentioned. If the studies have their own statistics which could be corresponded to your results do not hesitate to put the numbers to express the similarities and differences in the text. Also, references are needed in parts that you mentioned such comparisons.

Line 373 – Perhaps you could add more information about curriculum in PE here. Moral compass is often associated with group sport games such as volleyball, football etc. Norway athletes are strong in individual and winter sports. Maybe the results could be discussed from that angle.

You omit results of demographics in the discussion. Also, there is necessity to form an applicational value of this study. You could compare the Norwegian model of PE to other European countries based on raports like that

I have no relation to this publication, and I am not suggesting referring to this one. I am just putting an example of available data for comparison. Finding the differences in the curriculum or maybe grades from PE could lead to encouraging other countries to verify their model for the purpose of improving those indices that correlates the highest in your findings.

Line 405 – You reported my previous concerns here. I do not understand why you did not try to obtain more objective data as well. If there were ethical concerns here or private data management, please state it.

Limitations are well written with sufficient self criticism.

Line 448 – You address some of my previous concerns to line 373, but this description is too general. Without details about those conducted activities and comparison to other studies for example from U.S, your reasoning is limited.

To sum up, the study has potential to be published but authors need to put more emphasis on the details. Descriptive studies with questionaries could be discussed wider. Do not fear the bigger perspective or to wander about possible reasons of your outcomes if you put it in the sentences with proper amount of supporting evidence. Do not forget about the topic of the issue and focus more on it for making this paper more eligible for this special issue.

Author Response

Comments and Suggestions for Authors

Physical Education and its Importance to Physical Activity, Vegetable Consumption and Thriving in High School Students in Norway

 This study contributes to the body of knowledge about physical education and its benefits for the youngsters. This study findings are parallel to many other studies from all around the world, with standardised procedures and statistics. The main advantage of this study is well performed statistical analysis with up-to-date tools and computation. Authors have enough courage to properly state its limitations, but this does not justify lack of insight into the context of Norwegian curriculum on more specific level or the demographic context of the participants and their families. Methodologically, introduction needs to be rearranged to make the reasoning more straightforward and in order. Specific concerns are addressed below.

Comments

Abstract

Line 15 – Is M a mean age of participants?

RESPONSE: In the revision, it has been clarified that “M” means age. Please see Abstract.

Key values of statistical results could be put in numbers to highlight strength of evidence, especially for main topic, which is association between PE and Vegetable Consumption or Thirving.

RESPONSE: Thank you. The issue with key values of the statistical results is now addressed. Please see abstract.

Introduction

The quality of content and most of explanation are sufficient. What needs to be changed is a structure. The order of the introduction/background section is to introduce the reader to the topic, present premises which leads to certain assumptions. Then hypothesis is formed and aim of the research should be stated at the end of this section.

Present structure of:

Introduction

1.1. Physical education in the global and Norwegian contexts

1.2. Positive Youth Development, Healthy behaviors, and physical education

1.3. The Present Study

Should be rearranged to:

Introduction without stating the aim – just general presentation of topic with basic definitions. Context of PE in the world and Norway could be state here (from 1.1)

Clear description of PYD with exact explanation of each C. There is a place to do so, and reader will skip it than go to references with casual reading attitude.

Presentation of the current body of knowledge and its limitations, forming hypothesis and a reason to do this research, finally stating aim here.

RESPONSE: Thank you for the suggestion regarding the structure of the introduction. This has been considered in the revision.

 Methods

Line 205 – did the scores were self-reported or it was backed by their school grades and PE teacher evaluation?

RESPONSE: Yes, the scores of school grades were self-reported. This has been made clear in the revision and discussed as a limitation. Please see pages 5 and 13.

Line 217, Line 225 – if this is a result of this study, the information should be put into results section

RESPONSE: Thank you for the comment. The information about the Cronbach’s alphas is put in the results section.

Line 227 – Please add information that parent’s education level was put in the number scale accordingly from 1 to 5. There is also flaw here. There are 220 students. All students have two parents or caretakers in traditional heterosexual family? There is a chance that someone live only with one parent, or someone was adopted. Without clarification, this is a bias of this study and for group that large and randomly chosen students there is unlikely that all families present one model – maybe the lack of data in some places is due to this issue here.

RESPONSE: The number scale has been added to the respective categories. In the present paper, father and mother’s education were not analyzed as one parental educational variable but as two separate variables. They were also treated as control variables as they were not the main focus of the present study.

Line 239 – some of students were 16 years old. Is Norwegian law treating them as adults or consent ought to be obtained from their care takers/parents?

RESPONSE: In Norway, informed consent is not needed from the participants’ care takers/parents as long as the research, like the present study, is not dealing with sensitive themes, such as personal health issues and religion.

Line 241 – I understand that this external service performed language validation procedure. Do you have α-Cronbach values for respective items or other statistics that shows consistency with original version?

RESPONSE: Unfortunately, I do not have any findings that shows consistency with original version. However, the translation service used a double–checking procedure and translation experts in the relevant field of research that ensured the preservation of meaning.

Line 249 – Please state chosen demographic variables.

RESPONSE: These have been stated. Please see page 6.

Line 255 – How many students fulfil the data in 100% without necessity to fill in gaps? This will show exact extent of estimation that was performed.

RESPONE: The information regarding students with full data has now been added. See page 6.

Line 265-270 – Those hypotheses should be also placed in the introduction.

RESPONSE: The hypotheses in the introduction have been made clear now. See page 4.

Results

Please work on the aesthetics in terms of symmetry in all tables. For example, there is no need to split names in table 2 and column 1 in table 1 can be adjusted to one side or central.

Table 3 and 4 should be moved from discussion to result section.

RESPONSE: Issues with the tables have now been fixed. Please see Tables 1 and 2.

Discussion

Line 330-340 – This is repetition of results. In discussion you should try to compare results to a previous study that you mentioned. If the studies have their own statistics which could be corresponded to your results do not hesitate to put the numbers to express the similarities and differences in the text. Also, references are needed in parts that you mentioned such comparisons.

RESPONSE: Lines 330-340 was to provide a summary of the main findings of the study and to indicate whether the hypotheses formulated were confirmed or not. After that paragraph, findings are discussed considering earlier studies with references provided (indicated as numbers in brackets in line with the requirement of the journal).

Line 373 – Perhaps you could add more information about curriculum in PE here. Moral compass is often associated with group sport games such as volleyball, football etc. Norway athletes are strong in individual and winter sports. Maybe the results could be discussed from that angle.

RESPONSE: Thank you for this suggestion. I included some information about the curriculum but as “character” one of the 5Cs, which reflect a moral compass among other things, was not significantly related to PE grade I did not want to stretch the results.

You omit results of demographics in the discussion.

RESPONSE: Some results and discussion about the demographics has now been included. Please see pages 7 and 13.

Also, there is necessity to form an applicational value of this study. You could compare the Norwegian model of PE to other European countries based on raports like that

I have no relation to this publication, and I am not suggesting referring to this one. I am just putting an example of available data for comparison. Finding the differences in the curriculum or maybe grades from PE could lead to encouraging other countries to verify their model for the purpose of improving those indices that correlates the highest in your findings.

RESPONSE: Thank you for the suggestion. In the revision, some information on how the findings can encourage other European countries have been added. Please see the implication section.

Line 405 – You reported my previous concerns here. I do not understand why you did not try to obtain more objective data as well. If there were ethical concerns here or private data management, please state it.

RESPONSE: As the present study used a cross-sectional design, in line 405 I was trying to explain that the association between PE grade and youth development as well as the healthy behaviors could go both ways, and that longitudinal design can be utilized in future studies to investigate the role of PE sessions.

Limitations are well written with sufficient self criticism.

RESPONSE: Thank you for this comment.

Line 448 – You address some of my previous concerns to line 373, but this description is too general. Without details about those conducted activities and comparison to other studies for example from U.S, your reasoning is limited.

RESPONSE: The current findings have been discussed in light of earlier findings. Please see discussion section.

To sum up, the study has potential to be published but authors need to put more emphasis on the details. Descriptive studies with questionaries could be discussed wider. Do not fear the bigger perspective or to wander about possible reasons of your outcomes if you put it in the sentences with proper amount of supporting evidence. Do not forget about the topic of the issue and focus more on it for making this paper more eligible for this special issue.

RESPONSE: Thank you for your valuable comments. I have tried as much as possible to consider them in the revision.

Reviewer 3 Report

The manuscript entitled „Physical Education and its Importance to Physical Activity, Vegetable Consumption and Thriving in High School Students in Norway” presents interesting issue but some problems must be corrected.

Major:

The major problem associated with the presented study results from a small sample which was studied (only 200 students), which does not allow to conclude about the general population of students in Norway. Moreover, Authors did not verify representativeness of the studied group while compared with the general characteristics of the population of students in Norway, so we can not assesss the value of the results.

Abstract:

The aim of the study should be clearly formulated (e.g. “The aim of the study was…”).

Authors should present specific results of the study accompanied by the results of the statistical analysis.

Introduction:

Authors should reduce excessive information to present a brief justification of the study instead of everything that they know about the studied issue.

Authors should prepare this section not only to be interesting for Norwegian readers, but to be interesting for international readers. If Authors prepare their manuscript only for their national readers, they should publish it in some national journal. So, Authors should present here international data from various countries, not only the Norwegian ones.

Authors should properly justify their study – present what is so far known based on the other studies and what was not studied so far, to present that the conducted study may fill the gap in existing knowledge.

Authors should briefly formulate the aim of their study as it is in general presented in scientific articles (typically in the last paragraph, not in the first part of the section).

Materials and methods:

Authors should verify representativeness of the studied group

It seems that Authors did not verify normality of distribution

Authors should (1) verify the normality of distribution, (2) for normally distributed data present mean and SD values, but for the other distributions – present median, min and max values, (3) apply adequate statistical tests, that are based on the distribution.

Results:

Authors should present some descriptive tables describing the observed results within the studied group

It seems that Authors did not verify normality of distribution

Authors should (1) verify the normality of distribution, (2) for normally distributed data present mean and SD values, but for the other distributions – present median, min and max values, (3) apply adequate statistical tests, that are based on the distribution.

Discussion:

Authors must present a proper perspective for their results and should present more balanced opinions (see described above).

Authors should: (1) compare gathered data with the results by other authors, (2) formulate implications of the results of their study and studies by other authors, (3) formulate the future areas which should be studied.

Conclusions:

Authors should not present the second abstract, but direct conclusions (2-3 sentences to describe conclusions only)

Minor:

5Cs should be clearly defined while used for the first time

Authors should properly refer publications, e.g. instead of [see 28], it should be [28].

Tables are shabbily prepared and hard to follow (e.g. one table presented on 2 pages)

Author Response

Comments and Suggestions for Authors

The manuscript entitled „Physical Education and its Importance to Physical Activity, Vegetable Consumption and Thriving in High School Students in Norway” presents interesting issue but some problems must be corrected.

 Major:

The major problem associated with the presented study results from a small sample which was studied (only 200 students), which does not allow to conclude about the general population of students in Norway. Moreover, Authors did not verify representativeness of the studied group while compared with the general characteristics of the population of students in Norway, so we can not assesss the value of the results.

RESPONSE: Thank you for the comment. Under the limitation section, I have discussed how the current sample was quite small and may not represent the Norwegian youth population, so that future studies should consider a more representative study.

Abstract:

The aim of the study should be clearly formulated (e.g. “The aim of the study was…”).

Authors should present specific results of the study accompanied by the results of the statistical analysis.

RESPONSE: The aim of the study and well as specific results have been included in the revision.

Introduction:

Authors should reduce excessive information to present a brief justification of the study instead of everything that they know about the studied issue.

Authors should prepare this section not only to be interesting for Norwegian readers, but to be interesting for international readers. If Authors prepare their manuscript only for their national readers, they should publish it in some national journal. So, Authors should present here international data from various countries, not only the Norwegian ones.

RESPONSE: Thank you for the comment. The introduction section has been restructured to make clear both international and Norwegian perspectives on the topic of study. Please see section on introduction.

Authors should properly justify their study – present what is so far known based on the other studies and what was not studied so far, to present that the conducted study may fill the gap in existing knowledge.

RESPONSE: This comment has also been considered in the revision. See introduction section.

Authors should briefly formulate the aim of their study as it is in general presented in scientific articles (typically in the last paragraph, not in the first part of the section).

 RESPONSE: The aim of the study has been made clear in the revision. Please see page 4.

Materials and methods:

Authors should verify representativeness of the studied group

RESPONSE: The studied group is only a sample of the Norwegian youth population. Under limitation it has been discussed that the studied group may not be representative, something that future studies could consider.

It seems that Authors did not verify normality of distribution

Authors should (1) verify the normality of distribution, (2) for normally distributed data present mean and SD values, but for the other distributions – present median, min and max values, (3) apply adequate statistical tests, that are based on the distribution.

RESPONSE: Information stating that normality of distribution was tested has been included (page 6). In addition, the Mean and SD, as well as the min and max values (i.e., range) has already been provided for all study variables in Table 2.

Results:

Authors should present some descriptive tables describing the observed results within the studied group

It seems that Authors did not verify normality of distribution

Authors should (1) verify the normality of distribution, (2) for normally distributed data present mean and SD values, but for the other distributions – present median, min and max values, (3) apply adequate statistical tests, that are based on the distribution.

RESPONSE: Tables 1 and 2 are descriptive tables that were also summarized in the results section. In addition, some information stating that normality of distribution was tested has been provided. The Mean and SD, as well as the min and max values (i.e., range) have also been provided for all study variables in Table 2.

Discussion:

Authors must present a proper perspective for their results and should present more balanced opinions (see described above).

Authors should: (1) compare gathered data with the results by other authors, (2) formulate implications of the results of their study and studies by other authors, (3) formulate the future areas which should be studied.

RESPONSE: Under discussion, after the initial summary of the main findings and information regarding whether or not the formulated hypotheses were confirmed, the current findings were discussed in light of earlier research. Please see discussion section.

Conclusions:

Authors should not present the second abstract, but direct conclusions (2-3 sentences to describe conclusions only)

RESPONSE: The conclusion has been revised to make it more focused.

Minor:

5Cs should be clearly defined while used for the first time

Authors should properly refer publications, e.g. instead of [see 28], it should be [28].

Tables are shabbily prepared and hard to follow (e.g. one table presented on 2 pages)

RESPONSE: The minor comments regarding the 5Cs, the publication and the tables have also been considered in the revision. Thank you for your valuable comments.